# Towards Sustainable River Management of the Dutch Rhine River

**Hendrik Havinga**

Havinga River Dynamics, 6951BA Dieren, The Netherlands; hendrik.havinga@gmail.com

**Abstract:** Two thousand years of human interventions has heavily modified the Dutch Rhine river. Situated in a densely populated and developed delta, the river and its infrastructure fulfil important societal functions: safety against flooding, inland waterways, nature, freshwater supply, and agriculture. Programs to improve individual functions increasingly lead to conflicts with other functions and therefore call for an integrated approach. This paper reviews the history of the Dutch Rhine and documents the sectoral improvement programs in recent decades, explaining adverse effects such as the large-scale bed degradation at rates of up to 4 cm per year. The lessons from the past are used to propose avenues for future integrated and sustainable river training and river management, arguing that mitigating adverse effects while maintaining societal functions requires a combination of recurrent sediment management measures and extensive structural measures that may change the layout of the river system.

**Keywords:** river training; bed erosion; inland navigation; sediment management; longitudinal training walls; river management

---

## 1. Introduction

Two thousand years of human interventions have turned the Dutch Rhine river into a regulated river with dikes, groynes, low-flood levees ("summer dikes"), grasslands in the floodplains, and local nature areas. This layout reflects the consideration of interests associated with the river: flood protection, navigation, agriculture, and ecology. Recent programs to optimize these interests include the Room for the River flood control program, following the 1993 and 1995 floods and succeeded by the Delta Program for Rivers, and measures to increase biodiversity and improve the chemical quality of water in the European Water Framework Directive. Work is also underway to stop the ongoing bed erosion at rates of up to 4 cm per year, partly caused by the constrictions of a regulated river. Germany has already stopped its erosion; the Netherlands is working on preparations (Sustainable Fairway Rhine delta project). Measures that fit in a schedule to stop bed erosion are: longitudinal training walls, adaptation of groynes, fixed layers and riverbed nourishment by supplying sediment, accompanied by monitoring of measures and development of knowledge.

The river and its infrastructure fulfil important societal functions: safety against flooding, inland waterways, nature, freshwater supply, and agriculture. Programs to improve individual functions increasingly lead to conflicts with other functions and therefore call for an integrated approach. Closure of side channels and implementation of groynes in the 19th and 20th century profoundly changed the layout of the river. Today's increasing conflicts between different river-related interests will require major changes in layout again. Dredging alone does not help. The challenge is to develop a sophisticated set of structural measures, which is ecologically responsible and economically maintainable, even if effects of climate change become felt more and more. This is the basis for my thinking about future river management.

Major river improvement programs tend to be sectoral in the sense that they seek to optimize a single societal function or only a few functions, for instance safety against flooding and spatial quality (an amalgam of nature, landscape and cultural heritage) in Room for the River and ecological quality in projects for complying with the Water Framework Directive. These programs thus paid insufficient attention to their impact on inland navigation and river management.

The objective of this paper is to review experiences in order to formulate lessons learnt and to propose avenues for future river training and river management from an integrated perspective. Inevitably, this is colored by my personal experiences, working at the Rijkswaterstaat office of the Dutch Ministry of Infrastructure and Water Management (RWS) and abroad. It is time for a fourth Rhine river normalization in the Netherlands, consisting of structural adjustments to the wet infrastructure that will mitigate the negative impacts of the river training of the first, second and third normalizations in the 19th and 20th century as well as the negative impacts of recent measures with a Room for the River and a Water Framework Directive character. This paper does not provide new data, analyses or modelling results, but provides an integrated view, substantiated by references to more in-depth studies. This paper thus introduces river scientists and fellow river engineers and river managers to the peculiarities of the Dutch Rhine river.

## 2. Historical Development and Current Trends

### 2.1. Historical Development

This chapter provides a historical overview of the most important types of intervention in the alluvial parts of the Rhine river, focusing on developments in the Dutch Rhine delta and the German Niederrhein River. It has gradually become clear that the virtues of the heavily regulated Rhine system have not remained without serious disadvantages. The river training measures that have been introduced, such as dikes, width constrictions and bend cutoffs, made the river system lose its hydrological resilience. Before river training, the river could flood freely in flood plains and adapt its course when needed by cutting of bends, resulting in a rather constant geometry and no general bed erosion. In addition, the potential damage caused by future floods has increased (due to economic growth and greater population density), which can only be reduced by taking major measures and at high costs. Based on past experience, current river management strives to operate more in line with the natural behavior of the river. That is why the various Rhine states have started looking for technical solutions that can both maintain the original objectives and at the same time increase the hydrological resilience of the Rhine basin [1]. These solutions must be more flexible than traditional river training measures. Such goals can be achieved with the help of modern technology and increasing knowledge of the system. This new form of river management must recognize the natural dynamics and maximize its use. This chapter is partly based on [2].

In Roman times, dikes were built and creeks were dug to create the right conditions for agriculture on a local scale. Generals Corbulo and Drusus connected different river branches in the delta [3,4]. Since then, many river training measures have been implemented in the Dutch Rhine catchment area (Figure 1), resulting in a river landscape that is now completely different from the time of the Romans. Around the end of the first millennium, the population in Western Europe grew rapidly [4]. In the swampy soil of peat and clay, which at that time was 2 or 3 m above sea level, ditches were dug to lower the water level and make agriculture possible (between 900 and 1100 AD) The fall in the groundwater level caused oxidation of the peat, which in turn led to subsidence of the soil. After some time, this subsidence forced people to deepen the ditches and to dig canals to further lower water levels in order to maintain agricultural productivity. The permanent need to lower the groundwater level triggered an irreversible process of bottom subsidence.

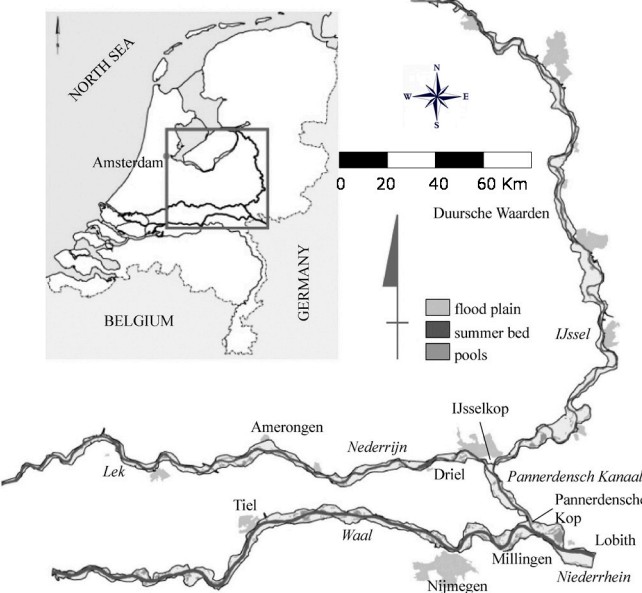

**Figure 1.** The Dutch Rhine branches (copyright RWS).

Around the year 1100, the subsidence was such that large areas adjacent to the sea were flooded during high tide. In addition to the man-made land subsidence, the natural rise in sea level also increased drainage problems. Both processes necessitated mitigating interventions, such as digging ditches, building dikes and dams, creating polders with artificial drainage, reclaiming wetlands, draining through so-called "boezems" on a large scale, and closing estuaries and inland seas. Dikes were built to protect the country against flooding. To prevent high water levels in the diked areas, the excess water was discharged via outlets at low tide.

However, the significant subsidence and the rise in sea level could not be stopped. The area behind the dikes and dams fell below mean sea level, making the drainage of the surplus water via gravity from the diked areas difficult and ultimately impossible. Small areas were diked behind the dikes and dams. From these small diked areas, better known as polders, the excess water was artificially removed and led to a former gulf or creek (around 1500 AD). The water was then discharged from these water courses via locks in the dam during ebb. The former coves and creeks were and are still used as water storage ("boezem") at high water levels. This step-by-step drainage system characterizes the polder landscape of the Netherlands.

Further, in the more upstream branches of the Rhine, the first river regulation works consisted of local dikes for flood protection. Groynes and dams were built along the river bed to prevent the erosion of the banks and to capture sediment to create agricultural land in the floodplains (situation a in Figure 2). Smaller channels were closed and many river bends were cut off. These measures were intended to increase the flow velocities in the main channel, thereby preventing the formation of sand bars. In winter, these shoals were sensitive to ice jams, which posed a serious threat to the dikes: they blocked flood discharges and mechanically damaged dikes. Later, these measures turned out to be favorable for navigation too, because they deepened the main channel. To further optimize the navigation channel, so-called width normalizations were carried out around 1870 (Figure 2). Width normalization is a form of river training by which the low water bed is limited to one channel with a constant (normal) width. Groynes were built at regular distances, so that during low discharges, the low water bed was restricted to a narrow channel, and the water flow was kept away from the erodible bank. Thus, three large-scale coordinated normalizations were implemented in the major Dutch Rhine branch, the Waal river, in the 19th and 20th century [3]. The first normalization took place between 1865 and 1879 (situation b in Figure 2) and the second normalization between 1879 and 1890 (situation c in Figure 2), narrowing the low water bed to 360 m. The third normalization took place between 1912

and 1916 and offered a low water bed width of 260 m, with a minimum sailing depth (LAD) of at least 2.5 m at low water levels (exceeded in 95% of time). This is still the main layout of the river today.

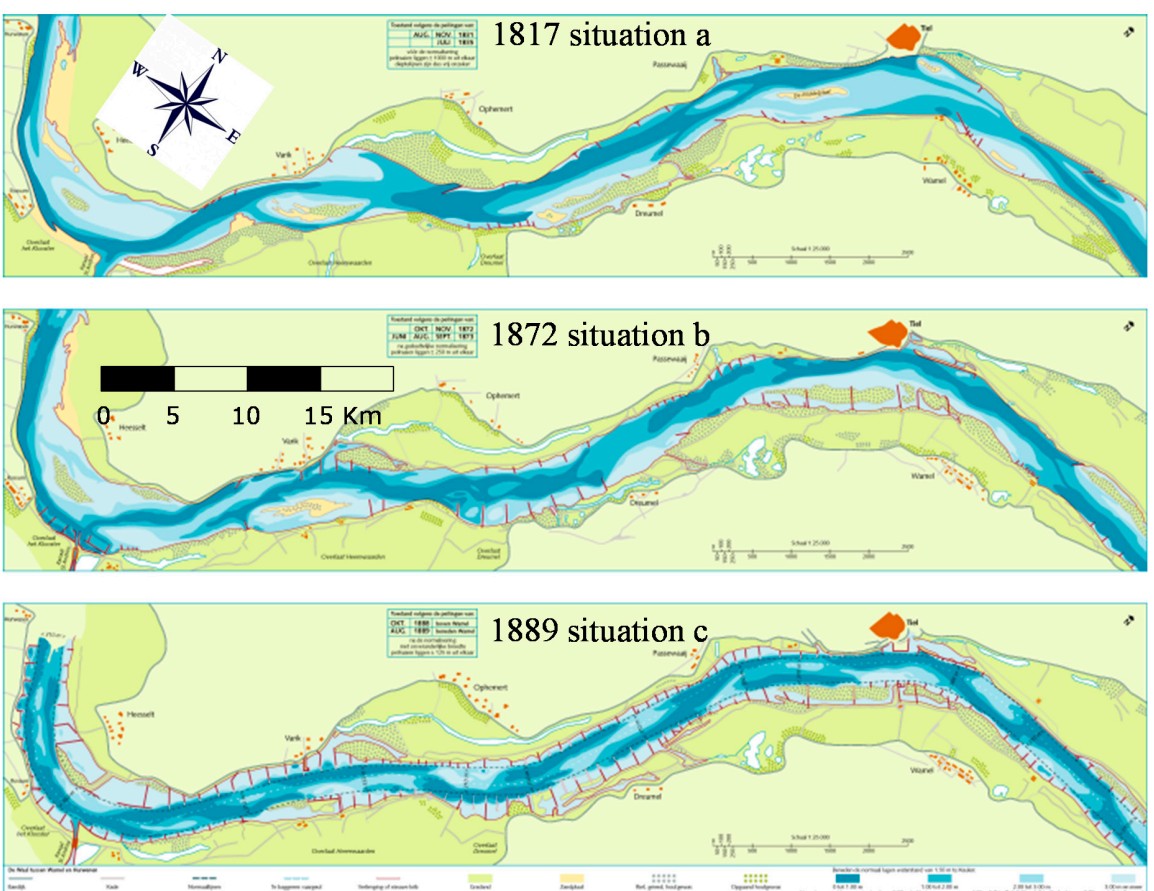

**Figure 2.** Stages in the normalization in the 19th century (copyright RWS).

In addition to normalization, stable river bifurcations were a condition for further social developments in the Dutch delta. The two bifurcations of the Rhine river in the Netherlands are the Pannerdensche Kop and IJsselkop (see Figure 3). These bifurcation points largely determine the amount of water that flows to the different Rhine branches. In the past, much attention has been paid to these bifurcation points in order to stabilize the flow and sediment distributions. The distribution of the Rhine water over the various Dutch Rhine branches caused many problems in the past. The upper reaches of the distributaries often branched in a crooked or perpendicular way off the main river branch. Such bifurcation geometries have contributed a lot to the historical poor discharge distributions [5] (see Figure 4).

The poor condition of the bifurcations not only caused flooding but also poor waterway transport conditions for the Nederrijn and the IJssel, which greatly affected navigation to the north (see Figure 4). After many discussions on conflicting interests, it was agreed to make an open waterway between the two rivers, the Pannerdensch Kanaal (see also [5]). It took 75 years before plans were approved to improve the mouth of the Pannerdensch Kanaal (see Figure 3).

From the end of the 18th century, the discharge distributions were stabilized, which was a prerequisite for comprehensive general river training of the Dutch Rhine branches. In 1745, the authorities of the Netherlands and Prussia agreed a water distribution, according to which the Waal river would discharge 2/3 of the total Rhine flow, the IJssel 1/9 and the Nederrijn 2/9 of the Rhine flow. Measures for flood defenses and navigation have been designed accordingly, meaning that above discharge distributions have not changed significantly.

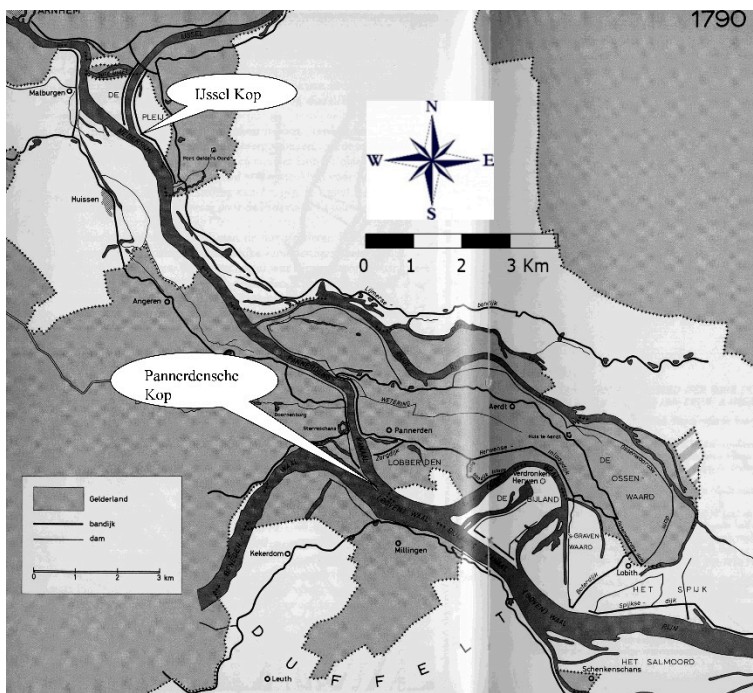

**Figure 3.** Rhine bifurcations, end of 18th century at Pannerden and Westervoort (copyright [5]).

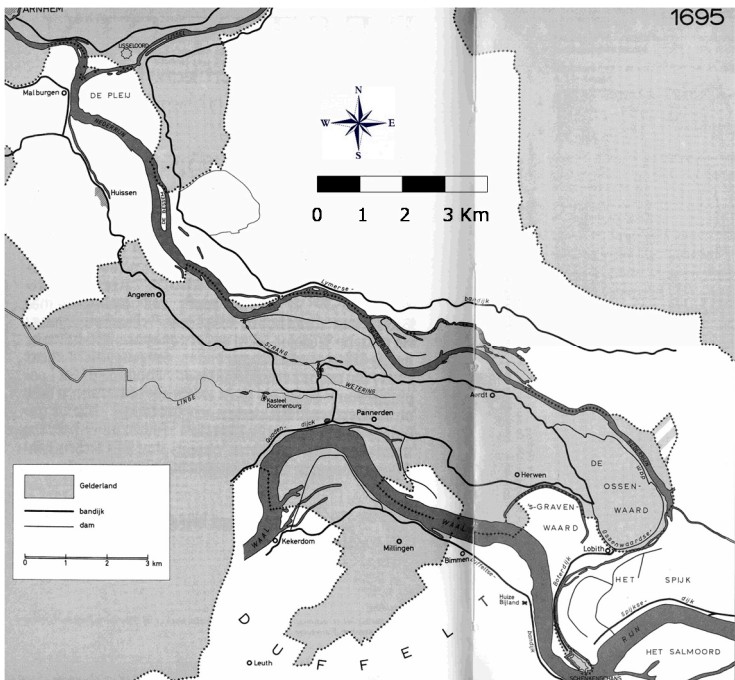

**Figure 4.** Rhine bifurcations, end of 17th century (copyright [5]).

The dikes and bend cutoffs entailed a significant loss of the original flood plains. The reduced river width causes a greater water depth with correspondingly higher flow rates. As a result of the reduced resistance to the flowing water, flood waves move faster and have higher peaks. Measures implemented in the German Rhine catchment between 1882 and 1955 caused the maximum flood peaks in Worms to increase from 5940 to 7760 m$^3$/s [6]. In anticipation of higher flood peaks, the dikes had to be regularly raised throughout the Rhine catchment area in order to maintain agreed levels of protection against flooding.

Interventions with specific goals often caused unexpected or underestimated hydromorphological reactions from the river system, which subsequently led to new measures to limit the reaction of the system. Table 1 summarizes the most important "intervention–response" relationships along the Dutch Rhine river.

**Table 1.** Important intervention–response relationships caused by river training interventions.

| Intervention | Hydromorphological Reaction | Mitigating Measure |
| --- | --- | --- |
| Drainage of peat areas in the Rhine delta | Oxidation of peat leads to subsidence on the land side of the dikes | Higher dikes and more powerful pumping stations in the polders |
| Regulation of river reaches and narrowing of the floodplains by dikes | Erosion of the river bed and lowering of the groundwater table. Higher flood peaks | Higher dikes |

These examples illustrate that almost every major intervention in the river system leads to reactions from the river, often also at a certain distance from the intervention (distant response in place and time). In the past, these responses were not always well understood. The net effect of all these interventions (including those along the smaller rivers in the Rhine basin) is a greatly reduced hydrological resilience of the Rhine river. Until the end of the 20th century, it was thought that the construction of even higher dikes and more powerful pumping stations would be the only control measure to be able to mitigate the effects of reduced hydrological resilience. Notwithstanding the benefits obtained, adverse effects are now recognized. They force us to reconsider the large-scale application of the measures described above.

The river bed of the Dutch Rhine branches has fallen sharply over the past two centuries, mainly due to the three normalizations. This bed degradation trend due to shortening and narrowing the river is the most challenging adverse effect.

In the German Niederrhein reach of the Rhine, structural measures and regular sediment supply have stopped this bed erosion in the last 4 decades. With a fixed layer at Spijk and sediment supply operations at Lobith, the Netherlands is also starting to actively limit bed erosion. The Rijkswaterstaat report MIT Verkenning Duurzame Vaardiepte Rijndelta [7] proposes alternative solutions to stop Dutch bed erosion. This stopping is desirable because in the foreseeable future the sailing depth would otherwise be reduced at the bed transition at the border with Germany where the bed has been stabilized (Figure 5).

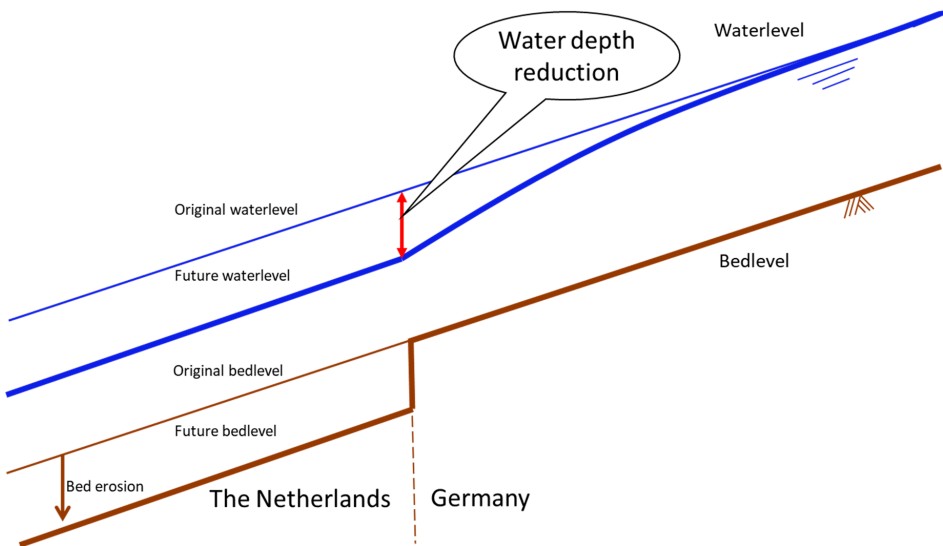

**Figure 5.** Reduction in water depth at the border due to downstream bed erosion.

The Netherlands already encounter a reduced sailing depth above non-erodible structures in the river bed (fixed layer in bend at Nijmegen, bendway weirs at Erlecom). Cables and pipes in the riverbed will also be exposed. The aforementioned report from 2007 already showed the continuation of bed erosion to be more expensive than taking (sustainable) measures. Moreover, continued bed degradation exacerbates the border crossing problem. Climate change will increase the problem because lower water levels are expected to occur more often.

*2.2. Current Trends*

Trends of change can be identified in climate change scenarios, affecting design discharges, as well as in discharge distributions at bifurcations, bifurcation geometries, bed levels, land use on floodplains and river banks, Rhine water temperature, and river management. Bed degradation at rates between 1 and 4 cm per year forms the major trend in bed-level changes (Figure 6) [7]. The Rhine water has become warmer due to regional warming and its use as cooling water for power plants. It is expected to become colder again due to the transition to renewable energy. This would increase the risk of ice jams. Guide walls have been built to avoid ice jams in the area of the bifurcations, but their functioning is not well understood. Nature restoration in the floodplains has increased the formation of shoals in the main channel and has rendered the behavior of the river less predictable and potentially more harmful to flood protection. River management in the last 30 years has seen a shift in focus from technical contents to managerial processes.

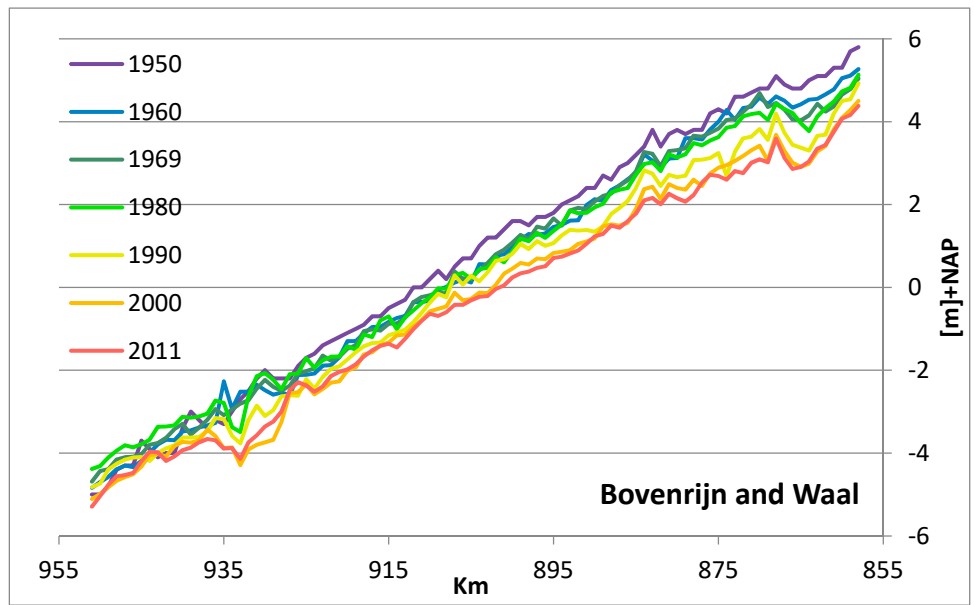

**Figure 6.** Average bed levels in Waal and Bovenrijn rivers between 1950 and 2011 (copyright RWS).

The shift in focus in river management implied a transition from solving problems by the Ministry of Infrastructure and Water Management (department of Rijkswaterstaat) itself to solving problems with the help of consultants, engineering firms and research institutes. Based on a desire of central control, many support services have also been centralized, implying that services regarding measurements, data collection and data processing are provided from a distance and that knowledge is concentrated at national organizational units. This has decreased operational knowledge within regional organizational components as well as the decisiveness in formulating problems and solutions that require these services. The work of government river engineers has taken on much more of a supervising nature without hands-on involvement. This has put the quality of river management under pressure.

### 3. Recent Sectoral River Improvement Programs

*3.1. Safety Against Flooding*

Floods in the Rhine river in December 1993 and January 1995 marked a turning point in the history of water and river management in the Rhine basin. A number of measures were identified to regain some of the hydrological resilience of the Rhine basin. Four important strategies, which deviate greatly from the traditional approach to water and river management, are:

1. The water storage capacity of urban and agricultural areas must be increased in order to prevent the rapid drainage of rainwater [2].
2. Further reduction in the space for the river must be prevented. Legislation called "Room for the River" is in force in the Netherlands. This will prevent construction activities in the floodplain that are not of crucial importance to society (such as housing in the floodplain).
3. The space available for the riverbed of the Rhine and its tributaries must be increased. This can be achieved by constructing retention polders and secondary channels, as well as removing obstacles in the river bed for the discharging water. The Room for the River (RfR) program [8] was implemented for this purpose between 2004 and 2018.
4. Nature values and biodiversity of banks and floodplains must be increased as much as possible when measures are implemented. This has been confirmed in the European WFD [9].

3.1.1. Prologue: Rhine in the Long Term

The Rhine in the long-term ("Rijn op Termijn") research project [10] showed that safe conveyance of a 25% higher design flood discharge (20,000 m$^3$/s) would require raising the dikes along the Dutch Rhine branches by more than one meter. As an alternative to raising dikes, this project proposes to channel the excess water through the IJssel branch to the relatively large IJsselmeer, which could serve as a temporary retention basin. This lake could be equipped with large pumps to transfer the excess water into the North Sea, if flushing by gravity) would prove to be insufficient. Reusing traditional measures, a bypass was proposed, which would only function during floods. Figure 7 shows how the IJssel river downstream of the new confluence would be considerably widened.

Retention basins on the land side of the dike are also useful for storing water to reduce the risk of flooding downstream. Figure 8 shows the Rijnstrangen area, located close to the Rhine bifurcation near Pannerden. Transformed into a retention polder, this area would fit well within a strategy of sustainable solutions for flood protection. Located in the upstream part of the Dutch Rhine branches, this retention polder, could reduce the peak water level of a design flood by 0.3 m, when opened at the right moment.

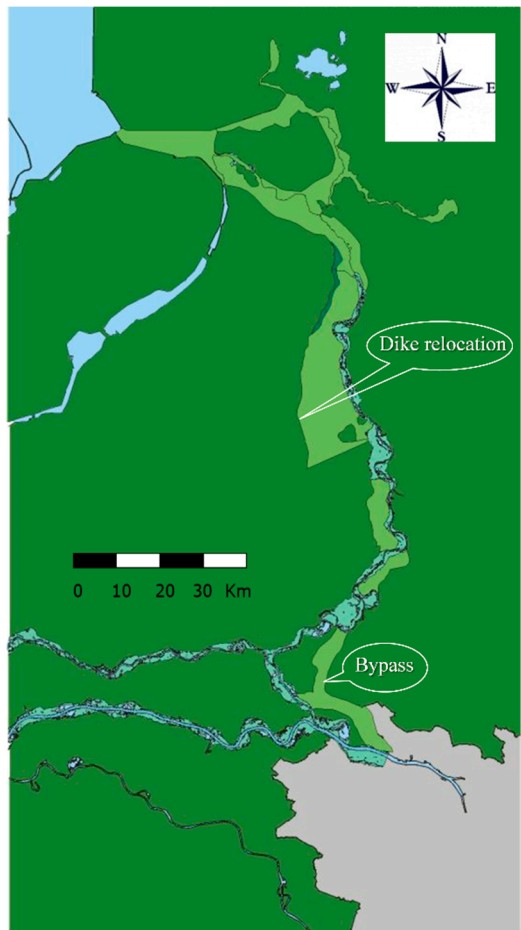

**Figure 7.** Bypass to the IJssel River and redevelopment of the IJssel (copyright [10]).

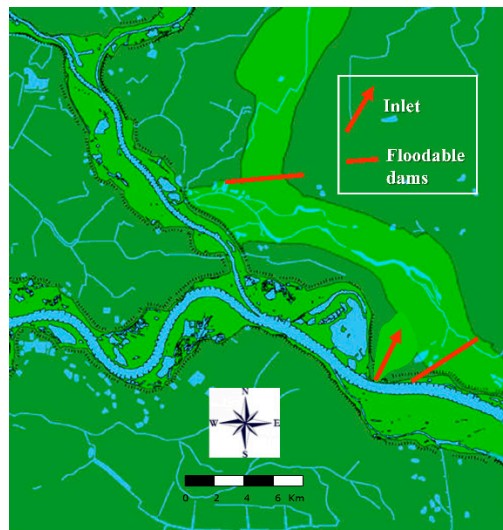

**Figure 8.** Retention polder in Rijnstrangen area (copyright [10]).

### 3.1.2. Room for the River

The aim of the Room for the River program (RfR, completed in 2018) was to increase safety against flooding without raising the dikes along the Rhine river and its distributaries. Raising dikes

as a continuation of traditional river management would increase the destructive power of a flood if a dike fails. The RfR program investigated and subsequently implemented: dike setback to widen floodplains, riverbed lowering, bypasses, secondary channels, removal of obstacles, groyne lowering and lowering of the floodplains that had aggraded as a result of increased sedimentation since the construction of the dikes (Figure 9). In addition, the program explored options for removing hydraulic bottlenecks to reduce the hydraulic resistance. This program offers solutions that are resilient and flexible, recognizing the need of flexibility in the face of an uncertain future under global change.

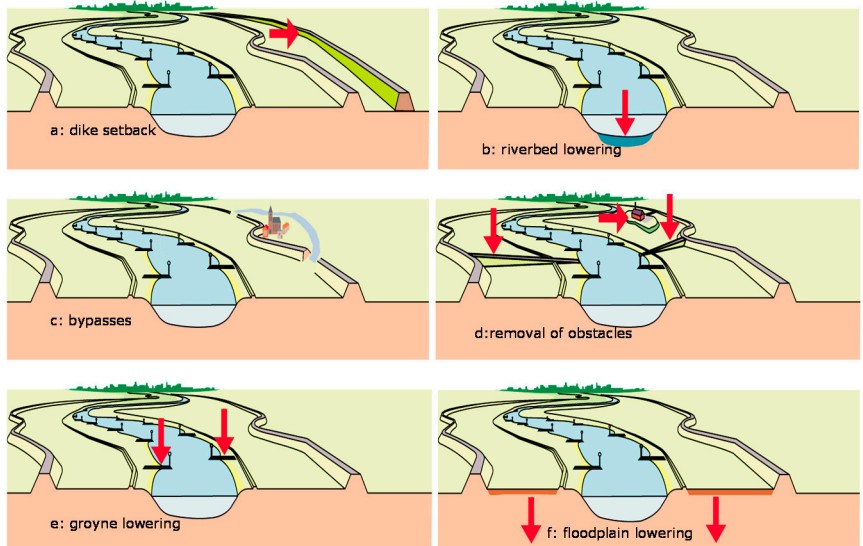

**Figure 9.** Measures of Room for the River. Top: dike setback to widen floodplains (**a**) and riverbed lowering (**b**). Centre: bypasses or secondary channels (**c**) and removal of obstacles or bottlenecks (**d**). Bottom: groyne lowering or replacement by longitudinal training walls (**e**) and floodplain lowering (**f**). (copyright RWS).

The works to increase the space for the river also aimed at enhancing spatial quality, an amalgam of nature, landscape and cultural heritage. Other interests such as navigation merely posed boundary conditions, accepting a certain increase in required maintenance. The package of measures increased the discharge capacity of the floodplain and the groyne fields, reducing the flows in the main channel. Transitions between modified and unmodified reaches generate alternating erosion and sedimentation during floods that travel downstream at lower flows. The resulting shoals increase the need of dredging for navigation. Natural vegetation in the floodplain must be controlled by active (cyclical) management in order to avoid an increase in flood levels (see [11,12]. Secondary channels require maintenance to guarantee the discharge capacity in accordance with their design.

### 3.1.3. Delta Program for Rivers (DPR)

The Delta Program for Rivers (DPR) is the successor to RfR and sets even higher flood safety targets. The measures consist of even more spatial (floodplain) measures and a few large bypasses (see Figure 10). Since the RfR measures can only safely guide a limited amount of extra water through the river, measures on the land side of the dikes should also be considered (e.g., bypasses and retention basins). Within the RfR program, measures were already implemented to meet the higher DPR standards: the bypasses at Nijmegen and Kampen.

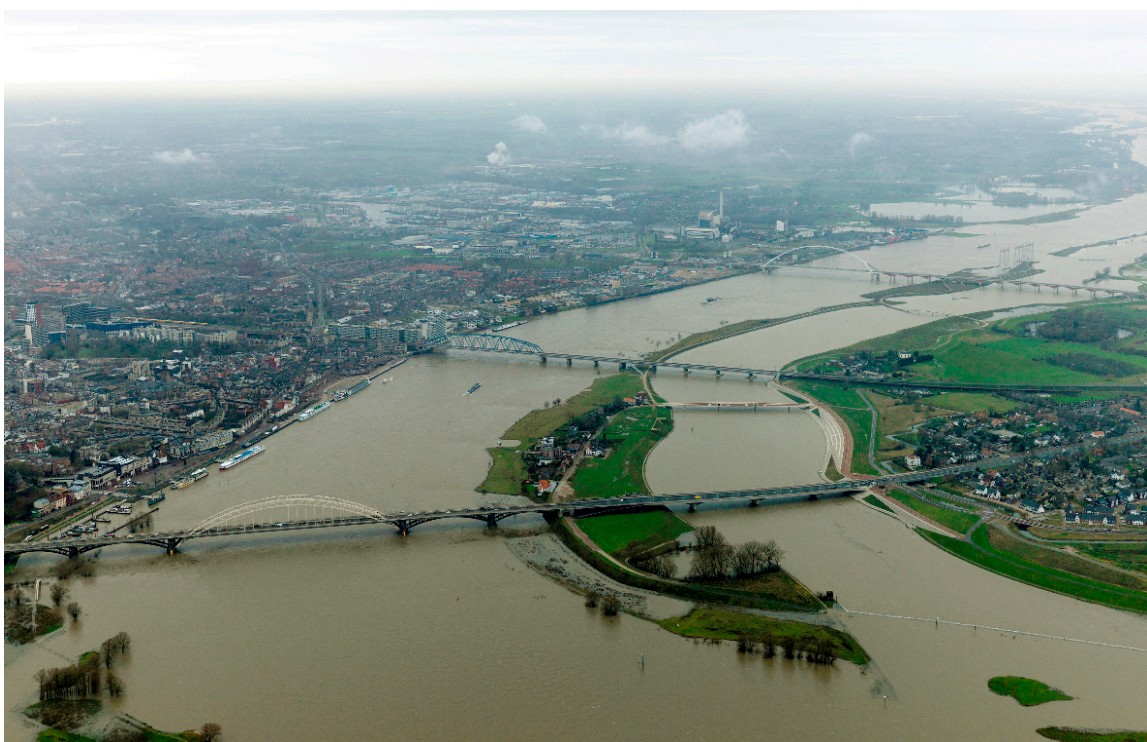

**Figure 10.** Bypass in the Waal river at Nijmegen (copyright RWS).

## 3.2. Inland Waterways

After the three normalizations in the 19th and 20th century, and the canalization of the Nederrijn (completed in 1970), the interest in navigation faded somewhat in favor of flood protection in the 1980s and 1990s. Already before 1993, political attention for inland waterway transport (IWT) returned, once again recognizing that IWT is an economically advantageous means of transport that also scores favorably ecologically. An idea to improve the waterway by cutting off bends between Nijmegen and the Pannerdensche Kop bifurcation (Figure 11) was abandoned in the 1980s due to problems with natural values and the consequences for the discharge distributions at the bifurcation points. Instead, bend improvement measures have been carried out (fixed layers and bendway weirs, see also Section 3.2.1).

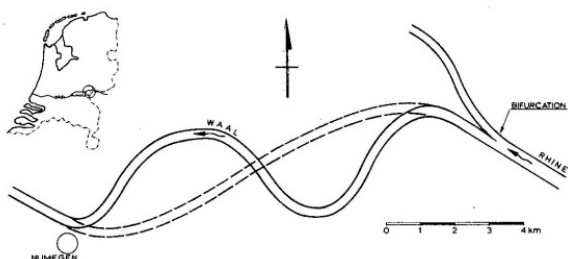

**Figure 11.** Proposed bend cutoffs in the Upper Waal.

The introduction of six-barge push tows in 1986 increased the pressure to improve the river as a waterway [13]. After a few studies on possible measures to improve navigation [14], it was decided to aim for an available navigation channel during Agreed Low Discharge (ALD) with a width of 150 m (Bovenrijn/Waal) and a depth of 2.8 m, previously 2.5 m. By definition, this ALD is exceeded 95% of the time. The accompanying target depth is called Least Available Depth (LAD). The related water levels are referred to as Agreed Low Water level (ALW). For the other Rhine branches, the depth at ALD remained 2.5 m, whereas the channel width varies per river section.

However, since the end of the last century, measures to improve IWT conditions have been under pressure internationally because navigation measures were linked to lower flood protection. An increased number of floodings along other rivers caused by floods and dike breaches since the 1990s fueled international commentaries on river works for IWT, already carried out or still in the pipeline. As a result, the Permanent International Commission for Navigation Congresses (PIANC) conducted research into the sustainability of waterways in relation to flood protection and ecology [15]. The research showed that measures for IWT do not have to lead to higher flood levels or worse conditions for nature development if certain conditions are met. The PIANC report discusses many examples of related projects on the Rhine and Mississippi rivers.

### 3.2.1. Bend Measures

After extensive scale-model research at Delft Hydraulics in the 1980s, a fixed (or 'armored') layer was chosen to improve the channel in the hitherto biggest bottleneck: the Nijmegen bend. The spiral flow naturally occurring in river bends (Figure 12) moves sediment from the outer bend to the inner bend. The outer bend becomes deep, even more than needed for navigation, but this depth is only available over a limited width because the inner bend becomes shallow.

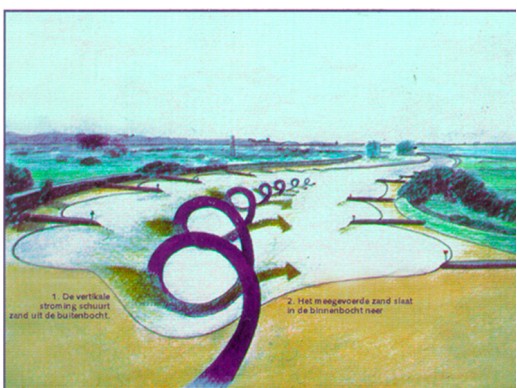

**Figure 12.** Artist impression of spiral flow in a bend (copyright Kees Nuijten).

The structural measure fixed (or armoured) layer in the Nijmegen bend consists of a bed raised with sand and gravel in the outer bend, that is covered with an erosion-resistant rockfill layer (equipped with a filter so that the sand does not pass through the layer). This reduces the cross-sectional profile, so that more water passes through the inner bend. The associated increased flow velocities erode the shallow inner bend while the spiral flow keeps the fixed layer free of sand (see Figure 13). Due to the increased flow rate and the smaller transverse profile, the water velocities in the inner bend are greater than in the original bend, so the sand can be transported through the bend. Downstream of the structure, the original transverse profile gradually reappears. At the transition, a scour hole is formed downstream of the sediment-free fixed layer, and a shoal in the same cross-section at the opposite side of the river where the faster flow from the inner bend decelerates (see Figure 14).

Fixed layers have been implemented in the Waal river at the Nijmegen bend (1985) and the St. Andries bend (1998). Figures 13 and 14 give an impression of the fixed layer. Figure 15 shows the location of the fixed layer St. Andries globally, and Figure 16 shows the obtained increase in the navigation channel width (at a depth of 2.5 m at ALW).

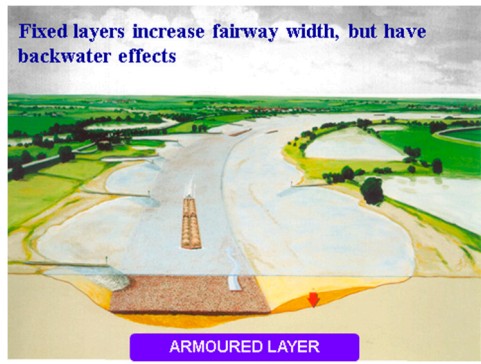

**Figure 13.** Fixed layer in outer bend (copyright Kees Nuijten).

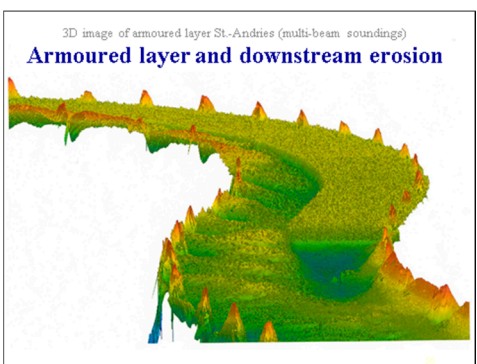

**Figure 14.** Morphological effects of fixed layer at the St. Andries bend (copyright Kees Nuijten).

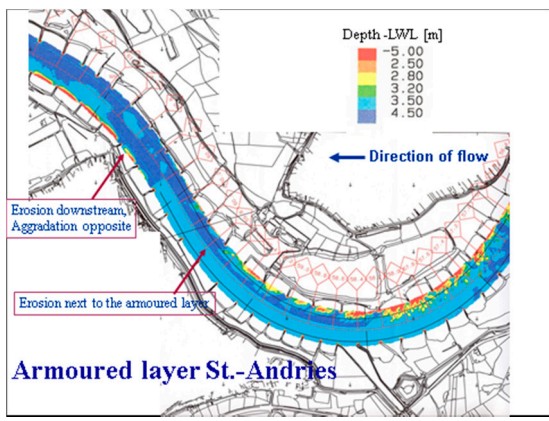

**Figure 15.** Location of the St. Andries fixed layer. Erosion in dark blue (copyright RWS).

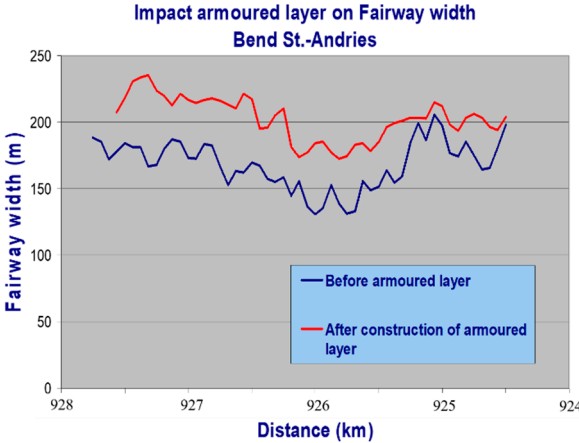

**Figure 16.** Increased navigation channel width at the St. Andries bend by fixed layer (at a depth of 2.5 m at ALW) (copyright RWS).

Bendway weirs instead of a fixed layer were used in the Erlecom bend (1996), because less width gain was needed there and less budget was available. Figures 17 and 18 give an impression of this measure.

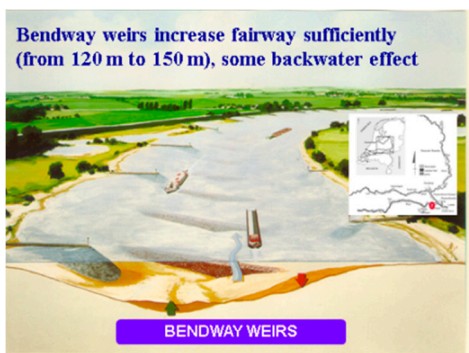

**Figure 17.** Bendway weirs in outer curve (copyright Kees Nuijten).

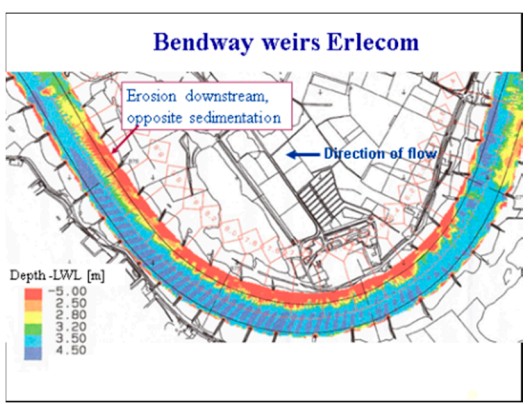

**Figure 18.** Location of bendway weirs at Erlecom (copyright RWS).

As an alternative to fixed layers and bendway weirs, research has been conducted on bend improvement using bottom vanes (Figure 19), experimentally [16] and theoretically [17,18]. The experiments showed that the bottom vanes had to be positioned at a certain angle with the flow for proper operation. Plans for implementation in the Waal bend at Hulhuizen were abandoned, however, when the research revealed that local erosion around the sheet piles constituting the vanes could

seriously reduce the intended hydraulic effect [19]. In my view, nonetheless, bottom vanes remain a potential solution for increasing the navigable width in bends. More information can be found in the aforementioned reports and in [20,21].

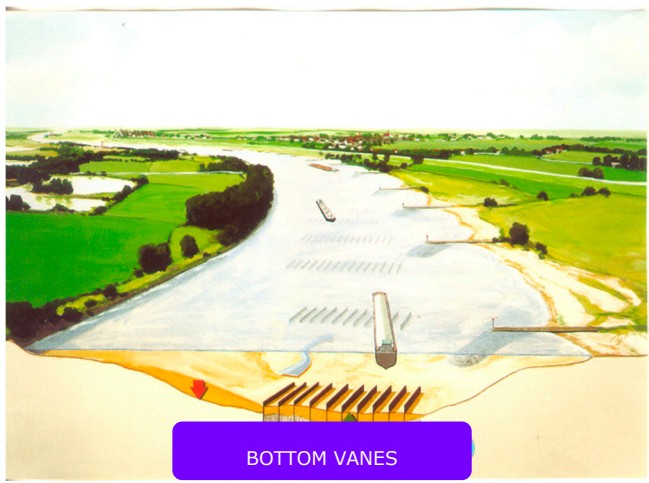

**Figure 19.** Impression of bottom vanes (copyright Kees Nuijten).

### 3.2.2. Dredging as a Permanent Measure

Under daily circumstances, with changing discharges and associated water levels, nautical bottlenecks are solved by dredging. This basic dredging demand is due to the occurrence of bed forms, dunes and ripples, and shoals due to flow patterns. Flow patterns can differ considerably during high discharge from the patterns during lower discharges, when all the water flows through the low water bed. Two developments in recent decades have led to more dredging to remove bottlenecks for navigation. First, the introduction of side channels and environmentally friendly banks, i.e., free eroding banks, since 1990 creates shoals at the inlet of the channels. Bank erosion products may deposit in the navigation channel too. Second, the decision in 2006 to aim for a minimum sailing depth (LAD) of 2.8 m at ALD on the Waal river instead of 2.5 m, without executing additional normalization works, necessitates continuous dredging for the 0.3 m extra navigable depth.

So, permanent dredging is required in order to maintain the agreed dimensions of the navigation channel. The dredged material is deposited in deep spots. Dredging contractors take care of the dredging in the context of a performance contract, that makes them responsible for the dimensions of the navigation channel.

In my opinion, dredging as a permanent measure is not sustainable and must be replaced by structural measures. Examples are extended groynes at secondary channels and longitudinal training walls. Though measures are an estimated 10-fold more expensive than the capitalized dredging, they provide a more stable waterway that navigation can rely on. Further, the mere presence of dredging vessels, hindering navigation, can be limited in this way. Figure 20 gives an impression of the dense traffic during low discharges in the Erlecom bend. Dredging equipment clearly limits the available sailing width here.

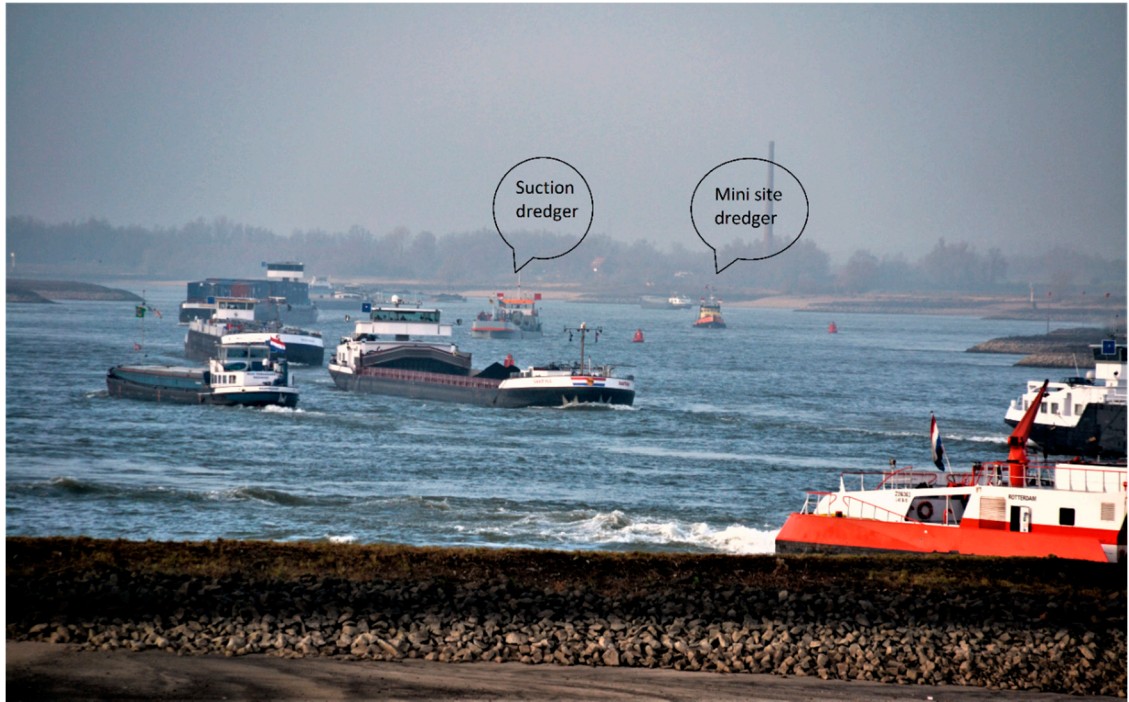

**Figure 20.** Traffic pressure at Erlecom during low water levels (November 2011).

### 3.2.3. Sustainable Fairway Rhine Delta

The objective of the concept of Sustainable Fairway Rhine Delta (SFR) is to obtain a sustainable inland waterway, taking into account climate change while stopping the ongoing trend of bed erosion due to river regulation interventions in the past centuries. In the German Niederrhein, this bed erosion has been stopped by implementing fixed bed structures and supplying sediment. In the Netherlands, bed erosion will have to be stopped too [7]. Otherwise, navigation will be hampered within 10 to 20 years, as a result of reduced sailing depth at the border transition (Figure 5) and above fixed bed structures (fixed layers and bendway weirs). Above fixed bed structures, already a decrease in sailing depth can be noticed during low discharges. The studies under SFR show that it is difficult to stop bed erosion while sustainably keeping the agreed navigation channel for the next 50 years [7]. Figure 21 shows one of the studied SFR strategies as an example (climate effect: 10% less discharge from Germany, no change in discharge distribution between the Waal and Pannerdensch Kanaal branches, no change in discharge to the Nederrijn).

This strategy includes longitudinal training walls to narrow the navigation channel at ALW, thereby increasing the depth for navigation. Longitudinal training walls divide the low water bed into a main channel for navigation and a bank channel. The sheltered bank channels offer more favorable conditions for riverine nature. Furthermore, secondary channels are no longer a problem for navigation if they are connected to the bank channel.

In 2014 and 2015, longitudinal training walls were constructed in a 10 km reach of the Waal river (Figure 22). In 2014, a fixed layer was constructed in the Bovenrijn bend at Spijk and in 2016 a first sediment supply scheme was put in operation at Lobith. These measures must slow down bed erosion, increase the channel depth and set up the water levels in upstream direction, in order to also reduce bed erosion in the German Niederrhein.

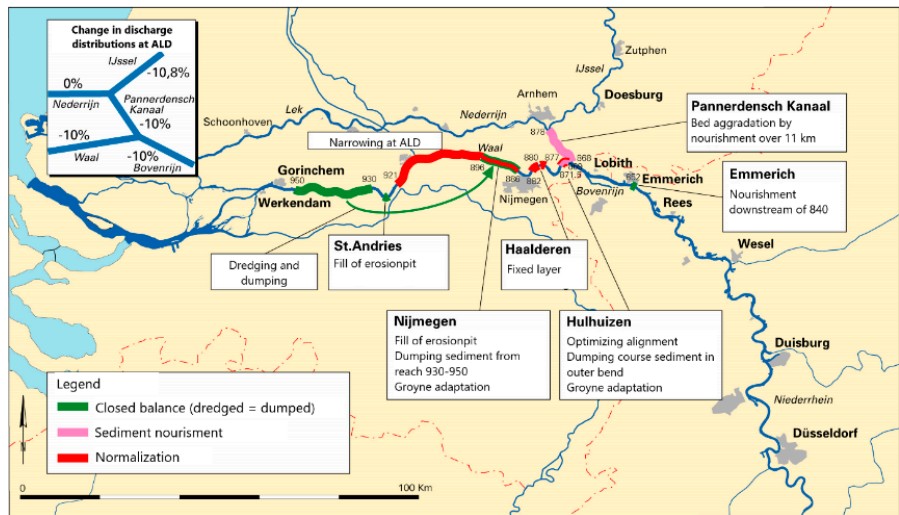

**Figure 21.** Sustainable Fairway Rhine Delta (SFR) solution direction B: compromise between water management and navigability (climate change alternative 10% less discharge, no change in discharge distribution at Pannerdensche Kop). Note the enormous impact on works to be implemented (copyright [7]).

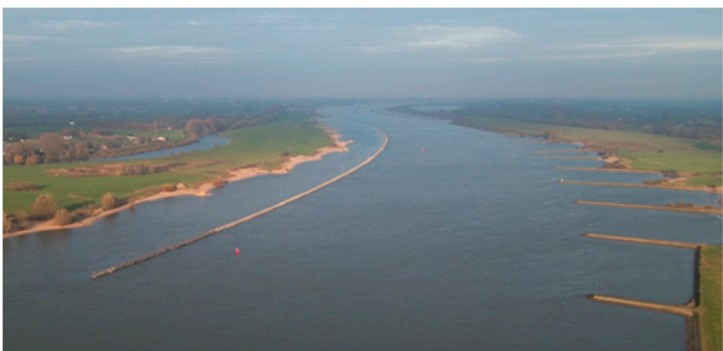

**Figure 22.** Longitudinal training wall near Ophemert (copyright RWS).

## 3.3. Nature

In order to increase biodiversity in the rivers and achieve a more natural landscape, several WFD nature redevelopment projects have been carried out and more river nature projects are in the pipeline. For ecological goals, the river needs to be managed differently than for the traditional interests of flood protection, navigation and agriculture (see also Table 2). Combining these interests with ecology requires in-depth studies into the morphological consequences of measures aimed at breaking the existing straightjacket of measures. In principle, ecological measures threaten the primary discharge function of the river and the navigation function in a regulated river, unless the intended vegetation and the associated intervention level can be tightly controlled.

By combining the effects of the existing river layout with the impact of proposed measures, the aggregated effect on hydraulics (design flood levels), morphology and navigation can be mapped. Experience shows that combinations of measures can be designed that increase the ecological potential without harming navigation. However, structural compensation measures such as groyne extensions or longitudinal training walls are often much more expensive than the proposed measures themselves. In practice, therefore, the recurring measure dredging and dumping of the dredged material is chosen to reduce hindrance to navigation, although it does not completely eliminate the hindrance. Extensive hydromorphological analyses are required to arrive at good measures designed in a sophisticated manner.

<div align="center"><b>Table 2.</b> Sectoral wishes with regard to design.</div>

| Function | Derived Goal | Geography | Defenses | Hydrology |
|---|---|---|---|---|
| Safety against flooding | Stability of flood defenses | Summer dikes Dikes Groynes | Stony structures Stone | Enforcement design flood levels and discharge distributions |
| Navigation | Maintaining channel dimensions | Summer dikes Large sailing depth | Stony structures Stone | Up to 4.0 m water depth: all discharge through main channel |
| Agriculture | Maintenance of area | Summer dikes Grass, no natural vegetation | Stony structures | No flooded floodplain |
| Ecology | More biodiversity through gentle land–water transitions | No summer dikes Secondary channels Natural vegetation | Soft defenses No stone | Often flooded floodplain |

Creating more room for the river by adding the river's former floodplains, or widening them, and restoring river habitats boil down to allowing more shoal formation in the main channel and more hydraulic resistance in the floodplains. Allowing natural processes necessitates a new approach to river management, referred to as Dynamic River Management (DRM). DRM prefers measures that are reversible and have no distant response. DRM acknowledges that the hydromorphological response of the river system by a control measure can only be partially predicted. So, to prevent irreversible effects, every new measure must be tested on a small scale. Only if the reactions of the system are positive (even after some time), scaling up of the measure can be considered. These measures tested on a small scale do not have major consequences, so that hydromorphological resilience is maintained. Instead of large rigid constructions, small-scale measures can be used to correct river reactions. These can also be measures with a renewable character, see for instance the Self-Supporting River System (SSRS) website at http://www.ssrs.info. Another important aspect of DRM is the involvement of stakeholders in the river basin. We learned along the Rhine river that implementing river management measures without public support is almost equivalent to mismanagement. For this reason, public participation has been a precondition for the preparation and implementation of the RfR program.

In practice, introducing DRM will lead to more dynamic changes in the river, both on the riverbed and in the floodplain, rendering river management more complex. The river manager must be prepared to take timely action, for which information about possible changes is essential. An extensive system of monitoring, control and impact assessment is required for this purpose (Dynamic River Management System or DRMS). It is laborious and expensive to adequately monitor changes that occur in the river bed and the floodplains with conventional techniques (aerial photos and field research). In recent decades, however, faster and cheaper techniques have become available (including drones), which are needed to monitor the size and structure of vegetation in large parts of a river basin (see also [2]).

The facts that river reactions are not always easy to foresee and that sometimes drastic changes in the use of the river system must be made lead to the conclusion that caution is required when altering the layout of the river system. Many measures must be developed that are not irrevocable, can be adjusted and do not cause system-wide responses.

Some standard navigation measures are cited to illustrate such measures. Due to their distant response, traditional normalization and bend cutoff measures can no longer be used to eliminate navigation bottlenecks. Instead, the waterway must be widened through locally effective measures. Examples of this are local bend measures such as fixed layers and bendway weirs, in addition to reducing the lateral exchange between the low water bed and the floodplain. The navigation channel is also maintained through dredging and dumping (a recursive measure).

The implementation of nature development projects increases the variability in the river area. A certain degree of vegetation development is permitted within the framework of these projects. The

vegetation must be controlled by cattle grazing, grass mowing, tree cutting, or complete clearance (with roots and all). Since the maximum hydraulic resistance is often only reached after 10 to 20 years and a wide variety of vegetation resistance may be expected, there is absolutely no certainty about the daily hydraulic resistance that individual nature development projects represent. Working with permits in which a certain maximum hydraulic resistance is included leads to the fact that hydrodynamic models (that include the permitted conditions) overestimate the hydraulic roughness, and hence the computed water levels. Therefore, in the current situation no good assessment of the computed design flood levels is obtained corresponding with the design flood discharge. In view of the major interests at stake, it is urgent that insight be gained into the current computed flood levels. A regular flood level assessment (e.g., a period of 1 year) is required for the day-to-day management and implementation of river projects for maintaining the agreed flood level criteria.

This motivates the need for a DRMS, able to regularly (roughly in accordance with the 4 seasons) display the current river situation with regard to design water levels (such as design flood levels and the ALW levels that serve as a reference for navigation at low discharges), discharge distributions and bed geometry of the navigation channel. In connection with the roughness caused by vegetation, this also includes a frequent update of the nature inventory. So, regularly updated field information is an integral part of a DRMS.

In summary, DRM has three objectives. First, the layout and management of the river area for the core tasks of flood risk management and inland navigation must provide the preconditions for restoring hydromorphological resilience: flood peaks and low-low water levels are softened, and bed erosion is stopped. Ecological recovery is taking part in this. Second, DRM strives for a sustainable river system, in which preferably small-scale measures are implemented that have no distant response. Third, the DRMS gives the river manager up-to-date insight into the physical and ecological status of the river, as well as into the requirements of the user functions. This insight is the basis for taking decisions about short-term and long-term measures.

DRM is therefore not a result, but a means (a management concept) to implement the policy regarding river user functions. DRMS is the operational system that is required for this and in which data management is central.

## 4. Implications for River Management

The RfR and WFD measures lead to aggradation of the low water bed and greater dynamics at the river bottom (see, e.g., Figure 23). In [8], a probabilistic analysis was carried out into the effect of the RfR measures on the Waal river. This showed that the effect on dredging operations is modest: some 50,000 m$^3$/y more dredging spoil compared to an average basic dredging need of approximately 400,000 m$^3$/y. However, the probability of a discharge hydrograph requiring more annual dredging than possible to maintain the agreed navigation channel, appears to be between 5 and 10% (dredging needs exceeding one million m$^3$/y). This element must be taken into account when choosing between dredging and structural measures.

Aggradation caused by RfR and WFD measures are on the order of 0.3 m, which is 8 to 11% of the available sailing depth. Although for each individual project the design is optimized to reduce the negative side effects for navigation, hindrance and economic damage to navigation cannot be prevented. This is because shoals are not prevented by adequate adaptation of the river layout (with groyne adjustments and bank protections). Shoals are merely removed temporarily by dredging. Main cause are the large costs of infrastructure measures and the relatively low costs of dredging.

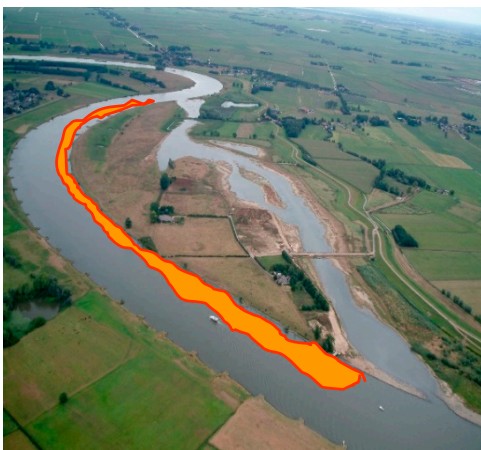

**Figure 23.** Possible aggradation through secondary channel Westenholte (yellow marking) (copyright RWS).

The maintenance costs for navigation, as a result of a combination of RfR and WFD measures, will nevertheless increase considerably as a result of the increase in the required dredging operations. The large amount of maintenance dredging that is required to keep the rivers navigable and the large number of dredgers needed will considerably hamper inland navigation, resulting in major economic effects and a lower safety of inland navigation.

It can be concluded that the recent changes in the balance of interests (in favor of ecology) have not yet been followed by a different layout of the river system, i.e., with structural training measures. System responses are only compensated by dredging. Consequences are barriers to navigation (hindrance, higher transport costs, lower safety), increased recurring maintenance (dredging), a potential decrease in safety against flooding, and an increase in $CO_2$ emissions. The best approach to limit dredging needs is to use more structural measures. The problem is that dredging is approximately 10-fold cheaper than structural measures. However, this comparison does not include all relevant parameters, such as an increase in transport costs due to local shoals, $CO_2$ emissions, and macro-economic shifts. If these effects are also valued financially, it will appear that dredging is less cheap and not economically sustainable.

The bottom line is that by implementing the aforementioned programs, river management becomes much more complex, expensive and intensive. If insufficient money and staff are available for management, monitoring, impact prediction and river training, an undesirable situation arises, namely less attractive inland navigation and a lower flood safety.

### 4.1. Limitation of the Morphological Effects

By systematically mapping the hydromorphological effects of all implemented measures, an estimate can be made of the effects on the morphology of the river bed (and river banks). This concerns local effects and cumulative effects of the various measures, that can move downstream and upstream. Morphological computer models can be used to calculate the bed levels for the coming decades. With this, the location, character and duration of bottlenecks in the navigation channel (small depths or widths) can be traced. By determining frequencies of occurrence of certain sailing depths, differences with the autonomous situation (without measures) can be determined. Depending on seriousness and character of system behavior and navigation bottlenecks, mitigating measures and dredging strategies can be developed, which are then tested in the same way. In addition to this way of analyzing implemented measures, specific research must be carried out to certain types of measures that are effective for as many interests as possible. This could include measures that are used in irrigation systems to control the sediment [22].

The bed level of a river determines flow depth and water level. These parameters are important for flood safety, navigation, ecology and agriculture. The bed level, however, is not a constant, but

depends on discharge hydrograph, sediment supply, sediment transport capacity and river geometry (including floodplains). Bed levels therefore change due to various interventions. The Rhine regulation in recent centuries has narrowed the river and thereby produced large-scale changes in bed levels and bed slope.

*4.2. Continued Bed Erosion*

Bed degradation at rates between 1 and 4 cm per year forms the major trend in bed-level changes (Figure 6). This figure also shows an estimate of the final equilibrium bed level [23]. The continuous bed erosion gradually gives rise to problems for the coverage of cables and pipes, the stability of structures (e.g., groynes) and the sailing depth (above structures, such as the fixed layer at Nijmegen).

4.2.1. Forecast of Bed Erosion

Different sources can be used to get an idea of bed degradation in the 'do-nothing' scenario. Changes in the bed levels have already been calculated in the survey on SFR (see [7]). At the fixed structures the bed of the Waal river does not degrade, but the surrounding bed does. After 30 years, the Waal bed level has lowered 0.3 and 0.7 m. The bed erosion causes a drop of the water levels, also upstream through backwater effects. So, where the bed cannot degrade, at the location of the fixed bottom structures, the depth is reduced by decimeters.

Table 3 from [24], which is partly the basis of [7], gives an estimate of the available sailing depth at locations with a non-erodible bed for a period of 30 years.

**Table 3.** Overview of depth above structures in navigation channel at ALD 2002 and ALD 2032 (copyright [24]).

| Location | km | Type of Measure | Depth at ALD 2002 [m] | Depth at ALD 2032 [m] | Difference [m] |
|---|---|---|---|---|---|
| Emmerich | 856 | non-erodible bed | 2.72 | 2.12 to 2.27 | 0.5 to 0.6 |
| Erlecom | 875 | bendway weirs | 3.64 | 3.19 | 0.45 |
| Nijmegen | 885 | fixed layer | 3.34 | 2.89 | 0.45 |
| St. Andries | 926.5 | fixed layer | 3.63 | 3.37 | 0.26 |

After 30 years, the depth reduction at ALD amounts to 0.5 to 0.6 m for Emmerich. Here, this problem has already been solved by replacing the natural non-erodible bed with a lower lying, artificial fixed layer. The depth reduction at Nijmegen is 0.45 m, and at St. Andries 0.26 cm.

In the Border Project [25], a 2D morphological model was used to analyze the ongoing bed erosion in the border area under the assumption that no bed erosion occurred in the Netherlands. Calculations showed a water level drop on the order of one meter on the Bovenrijn in 80 years (see Figure 24). Partly on the basis of this study, Germany has implemented bed-stabilizing works to counteract this drop in water levels (see the effect of sediment management at Rees in Figure 25). However, if the Netherlands does not implement similar works to support the water level, as described in [7], this fall in water levels will still occur. With a reduced waterdepth as a result.

In 2011, the bed-level changes on the Bovenrijn and Waal were calculated in [26] using a similar model. Table 4 shows the bed-level forecasts for 2015, 2050 and 2100 compared to the bed levels in 2010. These forecasts have been based on current river management and no implementation of additional measures to stop the bed erosion.

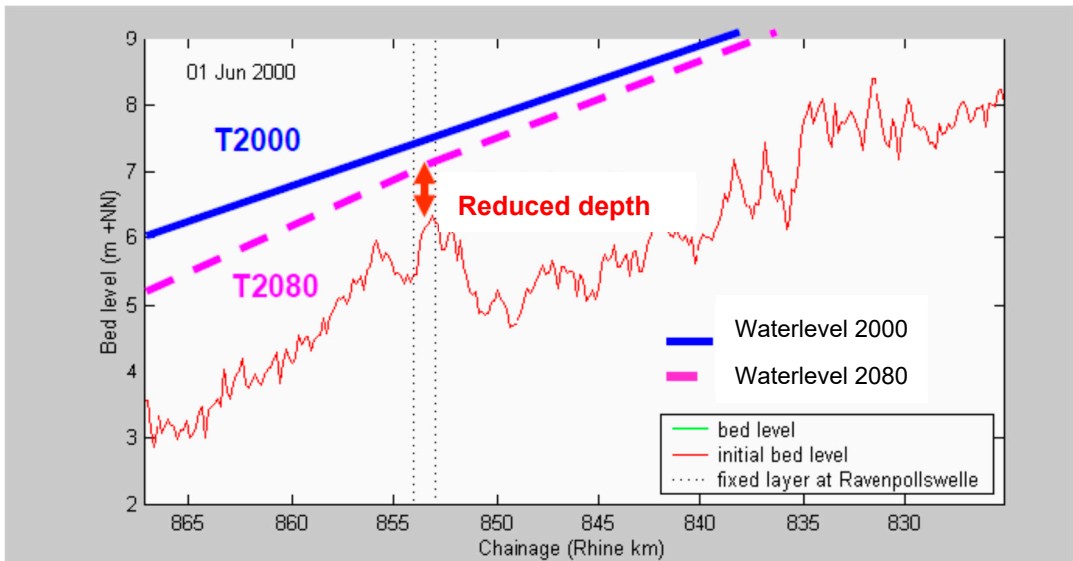

**Figure 24.** Water level effects as a result of the autonomous subsidence in the border area (copyright [25]).

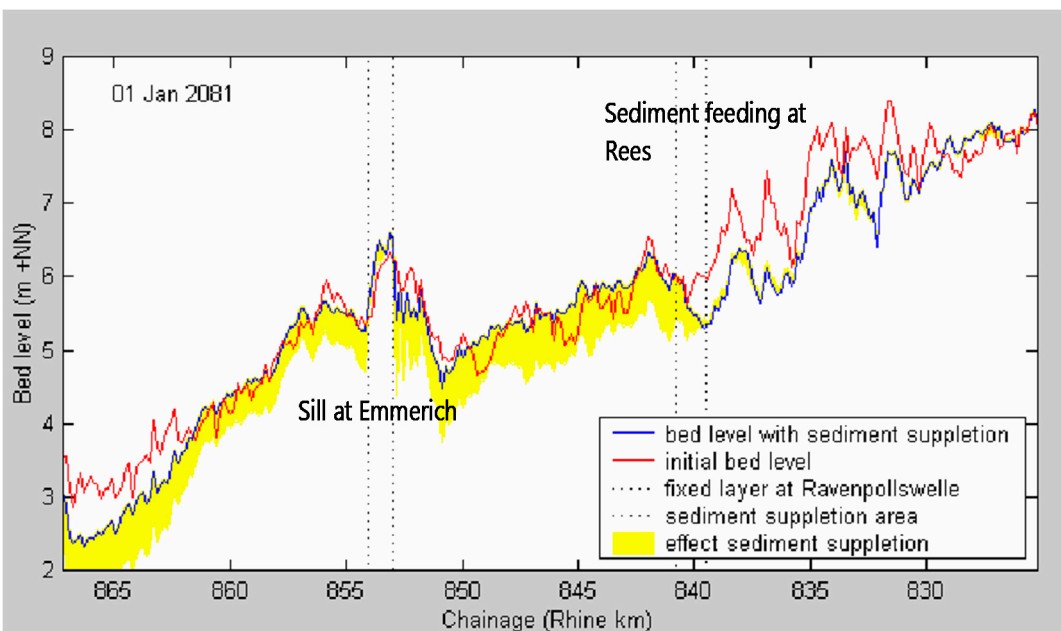

**Figure 25.** Bed degradation in 80 years and the effect of sediment supply in Rees (copyright [25]).

**Table 4.** Forecast of bed-level changes in Bovenrijn and Waal branches (copyright [26]).

| Branch | Period Location | 2010–2015 $\Delta z$ [m]/y | 2015–2050 $\Delta z$ [m]/y | 2050–2100 $\Delta z$ [m]/y | 2010–2015 $\Delta z$ w.r.t. 2010 [m] | 2015–2050 $\Delta z$ w.r.t. 2010 [m] | 2050–2100 $\Delta z$ w.r.t. 2010 [m] |
|---|---|---|---|---|---|---|---|
| Bovenrijn | Complete | −0.01 | −0.01 | −0.01 | −0.05 | −0.4 | −0.9 |
| Waal | Km 868 | −0.03 | −0.015 | −0.01 | −0.15 | −0.675 | −1.175 |
| Waal | Km 886 | −0.01 | −0.005 | −0.005 | −0.05 | −0.225 | −0.475 |
| Waal | Km 915 | −0.005 | −0.005 | −0.005 | −0.025 | −0.2 | −0.45 |
| Waal | Km 951 | 0 | 0 | 0 | 0 | 0 | 0 |

The information in Table 4 suggests that the bed erosion in the next 30 years will amount to 0.4 m on the Bovenrijn without interventions and approximately 0.2 m on the Waal above the fixed structures. This trend is expected to continue in the next 100 years.

### 4.2.2. Possible Measures to Control Bed Erosion

Analyses of WL|Delft Hydraulics [25] show that sediment supply alone does not work, but that sophisticated combinations of structural measures and sediment supplies are required to stop bed erosion and sustainably maintain the navigation channel (Figure 26). As a result of these studies, many structural measures have been implemented in the Niederrhein near Emmerich. The fixed layer in the Netherlands at Spijk (2014) and the sediment supply at Lobith (2016, 2019) have been logical follow ups and were also based on this study.

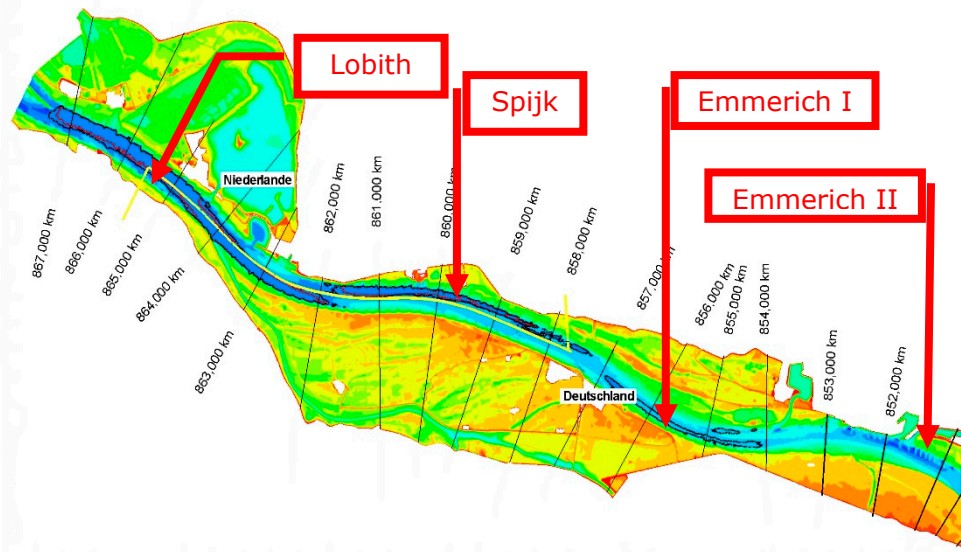

**Figure 26.** River bed stabilization and larger navigation channel by structural measures and sediment supply. Lobith: sediment supply and fixed layer. Spijk, Emmerich I and II: fixed layers (copyright [25]).

Currently, the recurrent measure of dredging and dumping is used to maintain the navigation channel. This measure cannot remove the reduction in depth at the border transition near Lobith. To what extent the success of this strategy is already limited by the fixed layers cannot be said without analyzing the LADs, but it is plausible that navigation already uses a narrower navigation channel in the bends in order to still offer the desired sailing depth. Despite the estimate that capitalized maintenance dredging is 10-fold cheaper than implementing structural measures, such as longitudinal training walls and groyne adjustments, in my opinion, dredging strategies are not a good alternative to structural measures to maintain the fairway in a sustainable way. Dredging and dumping operations do not yield a navigation channel with reliable dimensions. Moreover, dredging operations alone do not stop the ongoing bed erosion. It is then better to design the river training works required to stop bed erosion in such a way that the extra dredging work is minimized.

The transition problem near Lobith can be solved, but we should not wait another 10 years, because then there will no longer be a reasonable solution that will bring us back to the bed level of, say, the year 2000. The survey on SFR [7] provides three strategies for stopping bed erosion and sustainably maintaining the navigation channel. The corresponding measure packages consist of structural measures as well as groyne adjustments, longitudinal training walls, fixed layers and recurring sediment management operations (dredging and dumping). The recently implemented fixed layer at Spijk, the regular sediment supplies at Lobith and the longitudinal training walls in the Waal river reach Tiel−St. Andries fit in the SFR vision as these measures will slowdown the bed erosion and increase the navigation channel. The SFR2 report [24] provides a more recent state of affairs3.

### 4.2.3. Economic Effects of Continued Bed Erosion

Bosschieter [27] investigated the impact of climate changes for inland navigation, assuming a reduction of the sailing depth of 15 cm for ALD as a result of climate change in 2050 compared to 2003. The consequences are expressed in loss in load capacity. At ALD, the usable load capacity is 81% of the total load capacity of the fleet. For the fleet of class V motor vessels, only 67% can still be used at ALD and only 59% of the maximum load capacity for class V push tows.

The loss mainly occurs in the summer months when the lowest water levels occur. It amounts to a 4% decrease compared to 2003 [28]. Total loss in tonnes is estimated at 12 million tonnes.

The above numbers apply to a decrease in the sailing depth by an average of 15 cm. An indication of the effects of bed erosion on international transport can be obtained by multiplying the numbers by a factor of 4 for a depth reduction of 0.6 m as predicted in the morphological calculations for the Bovenrijn, yielding a total loss of 48 million tonnes (compared to the annual tonnage of 340 million tonnes in 2003). This means that an additional 48 barges per day (8 large push tows) would have to sail each day to compensate for the loss of load capacity. If the loss would have to be compensated by rail transport, 2000 extra train wagons would have to be deployed. These are an equivalent of 50 trains, approximately the entire capacity of the Betuwe railway. If the loss would have to be compensated by road transport, 5000 trucks would be needed. These trucks together have a length equal to 30 km (traffic jam).

Freight prices vary with the water level, between a few euros per tonne and more than €20 per tonne during low-water periods. An indication of the possible additional transport costs due to bed erosion can be obtained by assuming:

- Bed erosion of the Bovenrijn by 0.6 m in 30 years (estimated from Table 4);
- Loss of load capacity, especially in the summer months—freight charge of €20/tonne (marketprice);
- Loss of transport capacity (Rotterdam−Ruhr): 48 million tonnes (4 times 12 million tonnes loss per 15 cm loss of water depth);
- Loss of transport capacity of 12 million occurs once every 10 years [27];
- On average, there is a loss of transport capacity of 6 million tonnes per year if it is assumed that this loss is linear with the frequency of occurrence;
- For an erosion of 0.6 m, there is then a loss of transport capacity of approximately 24 million tonnes per year (4 times 6 million tonnes).

Ergo, the average extra transport costs due to bed erosion in the middle scenario amount to €480 million over 30 years. In a dry climate scenario, in which a year such as 2003 occurs every year instead of once every 10 years, the annual extra transport costs may amount to €960 million. There is no doubt that this will lead to a change in the modal split of the transport of goods, and thus more cargo will be transported by rail and in particular by truck, with known consequences for $CO_2$ emission, traffic congestion, and decreasing safety. Due to the cost increase, adverse macro-economic effects may also be expected because the position of the port of Rotterdam grows weaker and with that the competitive strength of the Netherlands.

In [7], the costs of maintaining the navigation channel in the Netherlands have been investigated in the event of *unrestricted bed erosion*. Continuation of the current daily river management (mainly dredging and dumping) cannot stop the bed erosion. This means that adjustments to infrastructure in the Netherlands and in Germany (such as lowering of river-crossing cables and pipes, structures and lowering of the fixed layers every 30 years) are needed with additional new structural measures. The cost estimate amounts to a minimum of €100 million in cash value (over 100 years with a discount rate of 5.5%). This is based on an additional annual (replacement) investment in the German waterway of €5 to €10 million—the present value of which is also €100 million. These extra costs for Germany would be caused by Dutch (mis)management and might lead to tension between the two countries.

As unchanged management will not stop bed erosion, another comparable amount will have to be spent in 30 to 50 years, whereby it is unlikely that further erosion will be accepted due to the

related problems (including replacement of navigation locks and excessive groundwater problems). The strategies to *stop bed erosion* cost more in the Netherlands, i.e., a cash value between €160 million and €200 million, but prevent investments in Germany. Today, even more far-reaching solutions with longitudinal training walls are considered, which may further increase costs.

*4.3. Sustainable River Management of the Dutch Rhine River*

Continuation of current river management, using mainly the recurrent measure of dredging and dumping, will not stop the ongoing bed erosion. The bed erosion is expected to continue for almost another century, leading to an extra bed lowering of approximately one meter. That will have an unacceptable negative impact on inland waterway transport, flood protection, ecology and agriculture. Analyses have shown that the cash cost of temporary measures (lowering of fixed layers, dredging and dumping) is on the same order as the cost of a combination of structural and recurrent measures: longitudinal training walls, adaptation of groynes, dredging and dumping. Nevertheless, the temporary measures do not stop bed erosion.

A recent study [29] addresses the problem of sustainable management of the river bed levels. Problems are acknowledged, but no hard conclusions are drawn. Short-term solutions are described, while sustainable long-term measures are only vaguely mentioned.

In my view, enough information is available today to design sustainable measures to stop bed erosion, even to let the river bed aggrade to historical levels. With current 2D morphological models and research outputs of the RiverCare program [30], steps should be taken to start implementing structural measures in all Rhine branches as soon as possible. The impact of the longitudinal training wall built in the Waal river is not clear yet. Nevertheless, solutions for the Waal river with longitudinal training walls at both sides of the river, with a sailing width of approximately 100 m (instead of the actual 260 m) and very wide bank channels behind the walls, should be investigated as an extreme showcase to stop the bed erosion and increase the sailing depth largely. This may lead to some hindrance to navigation, but ships will be able to sail at larger sailing depths.The adaptation of groynes in the upper Waal river reaches should also be investigated for the same reasons.

This fourth round of normalization works in the Rhine branches could efficiently mitigate the negative impact of recently executed works for improving flood protection and river ecology and may render river management more sustainable.

## 5. Conclusions and Recommendations

*5.1. Conclusions*

1. Historical development and recent sectoral improvement programs have shaped the Dutch Rhine river we see today. These recent programs include Room for the River, the Delta Program for Rivers, Sustainable Fairway Rhine Delta and the Water Framework Directive. They increasingly lead to conflicts between *safety against flooding and riverine nature rehabilitation*, thus calling for an *integrated approach*.

2. *Safety against flooding* has been a leading element in river management of the past two centuries. The stable discharge distributions established at the end of the 18th century are the basis for equal flood protection conditions along all Rhine branches. Recently, the level of flood protection has been increased by implementing the Room for the River program. Increased flood protection has been achieved by lowering floodplains, removing summer dikes, creating secondary channels, as well as removing obstacles from the river bed.

3. *Inland waterway transport* has benefitted from river regulation measures such as normalizations, widening the navigation channel in sharp bends and structural measures in bends, such as fixed layers and bendway weirs.

4. *Riverine nature* has been rehabilitated by implementing European Water Framework Directive measures such as side channels and natural banks.

5.  An *integrated approach* to river management is demanded, as the measures to achieve the goals of the Room for the River program and the European Water Framework Directive increase the discharge capacity of floodplains and groyne fields, resulting in reduced velocities in the low-water bed and therefore local aggradation. These local shoals hamper inland navigation. The situation is exacerbated by the continuous bed erosion due to the three normalizations. Problems arise with respect to the coverage of crossing cables and pipes, the stability of structures (e.g., groynes) and the sailing depth above structures.

6.  The estimated depth reduction in 2050 of 0.6 m on the Waal river results in a loss of tonnage of approximately 15%. In an average climate scenario, this leads to average extra transport costs of approximately €480 million a year. In a dry climate scenario, the annual extra transport costs may amount to €960 million. On the contrary, strategies to stop bed erosion are relatively cheap: in 2007, the costs were estimated to approximately €200 million. Today, more far-reaching solutions with longitudinal training walls are considered, which will further increase costs.

7.  The work of Dutch government river engineers responsible for river management has taken on much more of a controlling nature, which has put the quality of the products under pressure.

## 5.2. Recommendations

1.  A fourth normalization with an integrated approach is required, using structural measures to stop large-scale bed erosion. The measures must be implemented urgently, and can be designed with the help of current 2D morphological models and research outputs of the RiverCare program.

2.  This fourth round of normalization works in the Rhine branches could efficiently mitigate the negative impact of recently executed works for improving flood protection and river ecology and may render river management more sustainable.

3.  Solutions for the Waal river with longitudinal training walls at both sides of the river, with a sailing width of approximately 100 m (instead of the actual 260 m) and very wide bank channels behind the walls, should be investigated as an extreme showcase to stop the bed erosion and increase the sailing depth largely.

4.  Germany has already stabilized the eroding riverbed in the border reach of the Niederrhein river. In the Netherlands, stabilizing measures have been implemented only in the Dutch border region of the Bovenrijn river. To solve the transition problem, stabilizing works are demanded in the entire Waal river as well, to prevent the fall in water level at the border with a reduced waterdepth as a result.

5.  Geometric changes at bifurcation points have to be handled prudently because of possible effects on sand and water distributions, which in turn have effects on the most important user functions (flood protection, navigation, ecology and agriculture).

6.  A Dynamic River Management System should be implemented to control the impact of natural vegetation on actual flood water levels.

**Funding:** This research received no external funding.

**Acknowledgments:** I thank Erik Mosselman for inviting me to prepare this article and for his assistance in the laborious editing of the manuscript.

**Conflicts of Interest:** The author declares no conflict of interest.

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
