# Peer review of "Towards Sustainable River Management of the Dutch Rhine River"

_water, doi:10.3390/w12061827_

Round 1

Reviewer 1 Report

See review report

Author Response

I used abbreviations (e.g. WFD) in the main text, after a first explanation. I used the original tekst (e.g. Water Framework Directive) in the Conclusions and Recommendations section, as this section can be read without consulting the main text.

I tried to get the copyrights right, however with respect to the illustrator Kees Nuijten I found that he is no longer traceable on the internet (figures were made some 25 years ago). The figures without copyright have been made by myself.

Reviewer 2 Report

Review comments  on [Water] Manuscript ID: water-838042

This paper reported an overview of river management experience in Netherlands from multiple perspectives such as flood management, navigation, and ecosystem properties. Based on recognized challenges and issues in current and projective management, implications to river management was provided.

This is an important contribution to the freshwater science and management of rivers because it pointed out many issues commonly perceived in a variety of rivers across the world. Overall, it is well written. However, in places, there are redundant statements and they were present under different sections of the paper, which made it difficult to appreciate the content smoothly. Furthermore, although some terms such as RDM RDMS were discussed and comprises very important component of the paper, they were not well defined respectively when they first appeared. There were rather defined the of the sections. These are the major issues. Another possible improvement point is the addition of references to scientific evidence.

I left all comments on a separated submitted PDF. Please carefully revised.

Author Response

As you use a pdf, I commented within the PDF on your remarks.

I used abbreviations (e.g. WFD) in the main text, after a first explanation. I used the original tekst (e.g. Water Framework Directive) in the Conclusions and Recommendations section, as this section can be read without consulting the main text.

I tried to get the copyrights right, however with respect to the illustrator Kees Nuijten I found that he is no longer traceable on the internet (figures were made some 25 years ago).  The figures without copyright have been made by myself.

This manuscript is a resubmission of an earlier submission. The following is a list of the peer review reports and author responses from that submission.

Round 1

Reviewer 1 Report

See attached report

Reviewer 2 Report

The paper lays on an interesting research topic. It seems that the paper originates from a different format of a thesis or a technical report. It needs a full rewrite for further consideration. Your language command is not an academic. For example ". In my view, we must act more explicitly so as not to harm further the interests of the....... Here knowledge needs to be further developed, but river managers also have to act..". These mistakes are repeated. Without doing solid research, but based on a personal view, the research paper could bring a jargon to the scope of the subject area. 

Based on just secondary data, the author attempts to come to his/her conclusion.  

Reviewer 3 Report

I think that the topic is very interesting and personally, I will use it for my future academic and scientific work once it will be accepted.

Nonetheless, I consider that the manuscript shall be strongly modified and it cannot be considered for publication in its present form.

Below I list the general remarks regarding the manuscript:

The usage of English language is very informal. The author shall review the whole manuscript and avoid colloquial and informal phrases. Also, I recommend the usage of third person (e.g. the author thinks…) instead of first person (I think…). The quality of the figures shall be also improved. Some figures are not understandable, other lack of legends, other possess green background and they are not readable. Please try to standardize (as possible) the presentation of the figures. There is information that is unnecessary or redundant while there is lack of other information in other parts of the manuscript. I think that the manuscript is long and at the middle of it, it becomes tedious to read it. Most important: many of the statements are not completely supported by citations or by the results of the current research (if there is current research). It seems to be a manuscript with the personal opinion of the author regarding an important topic. The author should clarify the statements he gives.

Also, I can list some (not all) detailed comments:

Abstract: I recommend to rewrite it or at least to be more specific with the last part of this paragraph.

Line 23 erase the word in parenthesis (many)

Line 38 This is the basis for my thinking about… - avoid personal ideas (word thinking is also informal) and justify the statements you write. This situation is repeated in several parts of the manuscript.

Line 40 “in my view” same situation as above

Line 43 Unfortunately I am not from the Netherlands and I do not the Room for the River project and the Delta program… so some references shall be given

Fig. 2 Please change the green background, it becomes unreadable (instead of looking at the stages in water management, I was more interested in the yellow colour of the legends

Fig. 3 I think a citation is missed, also the legends are unreadable, I know the red lines are groynes, but not everyone knows it (e.g. first year BSc students).

Line 137 2/3rd or 2/3d? maybe the letters d and later th can be skipped

Figs. 4 and 5 it is difficult to appreciate the differences between them

Fig. 6 Please change the green and pink backgrounds, also increase the legends, which are the dimensions of the y axis?

Fig. 7 why is this big step in the river bed in the border between Germany and the Netherlands? Missing the classical triangle for representing the free water surface

Line 203 3. Autonomous developments: the historic developments were extensively described while this section is quite poorly described. Maybe the third chapter should be a subchapter of the chapter 2

Lines 218-223 strong words (maybe valid) but there is not a citation that supports this statement. This situation is repeated in many parts of the chapter 3, the facts seems to be evident, but they shall be supported by previous researching. Maybe, the author can auto-cite himself.

Fig. 8 Please make the legends more readable, why there are two question marks at the beginning and at the end of the green legend?

Fig. 9 There are two options, the author can remove the fig. of the left side and be more specific with the Fig. of the right side or the author can enclose the location of the bypass in the fig. of the left side.

Fig. 10 There is not Fig 10

Fig. 11 can the author explain this figure within the text? E.g. add a letter A to the first fig. and add this letter A in the line 314 – a) dike setback- and so on with the other 5 figs.  

Fig. 13 I consider this figure not necessary for the manuscript, or in case the author thinks it is, then it shall be better drawn. None of the figures indicate the flow direction, as I repeat, I am not from the Netherlands and it is for me difficult to visualize the description of the measurements the author describes.

Fig 13. There are two Figs 13, and also, I think that this fig. is not necessary.

Similar suggestion I can make until the end of the manuscript, regarding the figures and also with the formatting of the tables.

From chapter 5, the manuscript becomes clearer nonetheless, it is hard to follow all the aspects the authors threated. So, I strongly suggest to re-organized and shorten the manuscript, to add citations and to re-write the manuscript.

In general, I like very much the last part of the paper where the author is summarizing the manuscript. I just doubt that everything that it is in the last part was precisely described in the body of the paper (I come back to my statement that the article becomes very tedious to read).

I am open to review it again under the condition that the author will shorten, re-organize and improve the paper.

The manuscript is about a topic that is of general interest of the water engineering community. Indeed, I was excited on reviewing such interesting paper. Nonetheless, after reading each page, my excitement became lower and lower.

In general, the paper is interesting but it cannot be considered for publication in its actual form. I think that the author expressed his point of view (very valid) on how the Dutch part of the Rhine shall be managed.  The author also states very useful and valid conclusions and recommendations, but from my particular point of view, those statements need to be better justified.

Also, the English language is correct but in many parts is very colloquial and informal. It seems to be more an article for a popular science newspaper such as national geographic, than for a peer review scientific newspaper.

I am open to review it again under the condition that the author will shorten and improve the paper.